# Electric Drivetrain Optimization for a Commercial Fleet with Different Degrees of Electrical Machine Commonality

**Meng Lu [1], Gabriel Domingues-Olavarría [1,2], Francisco J. Márquez-Fernández [1,*], Pontus Fyhr [1,3] and Mats Alaküla [1]**

[1] Division of Industrial Electrical Engineering and Automation, Lund University, SE-22100 Lund, Sweden; meng.lu@iea.lth.se (M.L.); gabriel.domingues@iea.lth.se (G.D.-O.); pontus.fyhr@iea.lth.se (P.F.); mats.alakula@iea.lth.se (M.A.)

[2] BorgWarner Sweden AB, SE-26151 Landskrona, Sweden

[3] Haldex Brakes AB, SE-26124 Landskrona, Sweden

[*] Correspondence: fran.marquez@iea.lth.se

**Abstract:** At present, the prevalence of electric vehicles is increasing continuously. In particular, there are promising applications for commercial vehicles transferring from conventional to full electric, due to lower operating costs and stricter emission regulations. Thus, cost analysis from the fleet perspective becomes important. The study of cost competitiveness of different drivetrain designs is necessary to evaluate the fleet cost variance for different degrees of electrical machine commonality. This paper presents a methodology to find a preliminary powertrain design that minimizes the Total Cost of Ownership (TCO) for an entire fleet of electric commercial vehicles while fulfilling the performance requirements of each vehicle type. This methodology is based on scalable electric machine models, and particle swarm is used as the main optimization algorithm. The results show that the total cost penalty incurred when sharing the same electrical machine is small, therefore, there is a cost saving potential in higher degrees of electrical machine commonality.

**Keywords:** fleet optimization; electric commercial vehicles; total cost of ownership; electrical machine scaling





## 1. Introduction

Electric commercial vehicle fleets can be economical compared with conventional commercial vehicle fleets based on fossil fuels [1]. Particularly in Sweden, they benefit from the relatively high price of fuel and low price of electricity. Additionally, the Swedish electricity mix has an exceptionally low carbon content [2], further reducing the environmental impact of commercial vehicles.

For electric vehicles in general, and even more so for heavy duty commercial vehicles, the battery is the single most expensive component. However, with the development of battery technology and battery manufacturing processes, the benefits of economies of scale, as well as the potential mass-deployment of charging infrastructure allowing the vehicles to operate with a smaller battery capacity (e.g., a dense fast charging network or even Electric Road Systems for dynamic charging [3]), the relative importance of the battery in the overall upfront cost decreases, hence the powertrain plays a more important role in the cost analysis. The upfront cost of the powertrain may be reduced by finding commonalities between different vehicles in the fleet, although this could negatively affect operation cost due to reduced efficiency.

This paper assesses the impact of different degrees of powertrain commonality on the total cost for the whole fleet by finding the powertrain configuration that minimizes the total cost (upfront cost of the powertrain and operating cost of the vehicle) for each of the cases.

For electrified vehicles, the powertrain includes the power electronics converter (PEC), the electrical machine (EM) and the mechanical transmission (MT). In order to analyze the cost of the powertrain, accurate and scalable models of the different components, especially the EM, are needed to create a large enough design space as input to the optimization process.

Several methods are proposed in the literature to improve the scalability of EM models. The usual practice is to generate a "base" machine in a two-dimensional finite-element analysis (FEA) tool and then new machine designs can be generated with limited computational time by changing the size of the machine and number of turns of the windings [4]. Often, the change in the thermal performance due to geometrical changes is neglected during the scaling process in this method, so a thermal lumped parameter model for the EM, as described in [5], can be used to take into consideration the end winding region and also to evaluate the overloading capabilities for the scaled EMs.

There are also several papers trying to break down the cost of powertrains to the cost of the components, such as the type of power semiconductors used in the PEC or the gears in the MT, to analyze the impact of different design parameters on the cost [6,7]. In [8] a model to estimate the cost of the powertrain by a fixed material price and an empirical formula is proposed, resulting in low computational effort models.

The definition of the different cases considered regarding the degree of commonality and the specificities of the fleet are given in Section 2. Section 3 describes the performance and cost models used for each of the components, as well as the overall structure of the optimization scheme. Section 4 compiles the most relevant results obtained from the study, and these are discussed further in Section 5.

## 2. Problem Description

### 2.1. Degree of Powertrain Commonality

Assuming a certain fleet of commercial vehicles, with known characteristics (weight, air drag coefficient, continuous and max. power rating, etc.) and operating conditions (representative drive cycles), the first step is to divide them into as many groups as the desired number of electrical machine geometries. Each of the groups then contains one or several different vehicles, with one or several representative drive cycles each.

In this article, three different cases are considered: (1) each vehicle type in the group has its own optimized EM, (2) all vehicle types share the same EM 2D geometry but have individually optimized axial lengths and (3) all vehicles in the group feature the exact same EM geometry but keep the same PEC and MT as in the first case. The winding configuration, the PEC and the MT are tailored to each vehicle type in the group.

### 2.2. Fleet Specifications

The fleet of electric commercial vehicles considered in this article aims to represent the existing fleet in Sweden [9]. It consists of nine types of vehicles, varying from city distribution trucks with 11 tons to long haul trucks with 35 tons. The number of trucks of each type, as well as the annual driving distance is known, which can be seen in Table 1.

The performance of a truck is defined by its top speed, gradeability and cruising speed with certain slopes. These requirements define the wheel side torque envelopes which will later determine the gear ratios of the mechanical transmission for a certain electrical machine. The fleet information and vehicle requirements can also be seen in Table 1.

The three drive cycles used to evaluate the energy consumption of each vehicle type are shown in Figure 1.

**Table 1.** Fleet information and performance requirements for the fleet.

| Applications | | Weight (ton) | Number of Trucks | Annual Mileage [km] | Power Requirement Cont./Peak (kW) | Applied Drive Cycle |
|---|---|---|---|---|---|---|
| Group A | Vehicle 1 | 11 | 6107 | 15,910 | 108/162 | City |
| | Vehicle 2 | 14 | 6365 | 16,810 | 114/171 | Distribution |
| | Vehicle 3 | 18 | 13,588 | 29,300 | 132/198 | drive cycle |
| Group B | Vehicle 4 | 21 | 2631 | 28,340 | 204/306 | |
| | Vehicle 5 | 23 | 825 | 10,900 | 216/324 | Regional |
| | Vehicle 6 | 25 | 6933 | 35,270 | 228/342 | Distribution |
| | Vehicle 7 | 27 | 31,811 | 64,450 | 234/348 | drive cycle |
| | Vehicle 8 | 29 | 8657 | 63,260 | 240/360 | |
| Group C | Vehicle 9 | 35 | 11,524 | 44,730 | 264/396 | Long-Haul drive cycle |
| **Performance requirements for the fleet** | | | | | | |
| Top speed | | | | 120 km/h | | |
| Gradeability | | | | 50 km/h @ 5% 90 km/h @ 1% | | |
| Startability | | | | 16% | | |

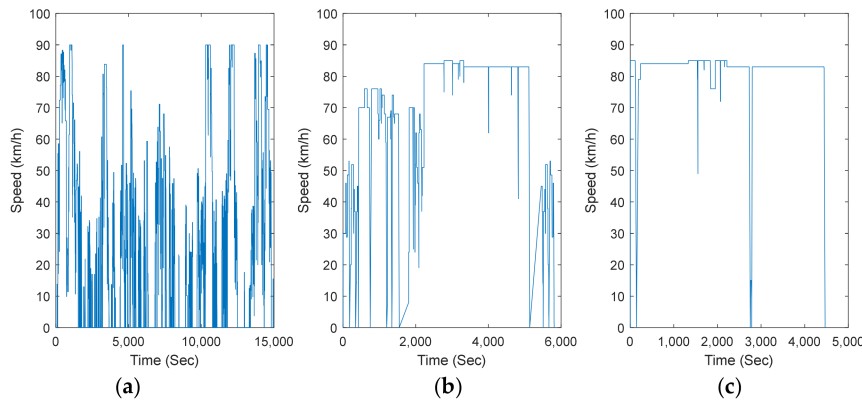

**Figure 1.** (**a**) City Distribution drive cycle; (**b**) Regional Distribution drive cycle; (**c**) Long-Haul drive cycle.

## 3. Methodology

### 3.1. Optimization Procedure

The optimization process that follows for each of the groups is independent, and is based on the work presented in [6]. Every EM geometry (2D) in a previously defined set will be evaluated, and the optimizer will find the optimal machine length and number of turns, together with the PEC rating and the MT to minimize a certain objective function, for each of the vehicle types in the group. Once all machine geometries in the set are evaluated, the one with the lowest total objective function (adding the results obtained for all vehicles) is selected as the best machine for that group. The optimization process for one group is shown in Figure 2, and it is repeated for all groups.

### 3.2. Objective Function

The objective function to be optimized is the sum of the upfront cost of the powertrain depreciated over 10 years and the annual operating cost. The upfront cost of the powertrain consists of the cost of EM, MT and PEC. Other components such as control electronics, pumps and fans are not included in the calculation.

The corresponding cost models for both upfront and operating costs are presented in the next sections.

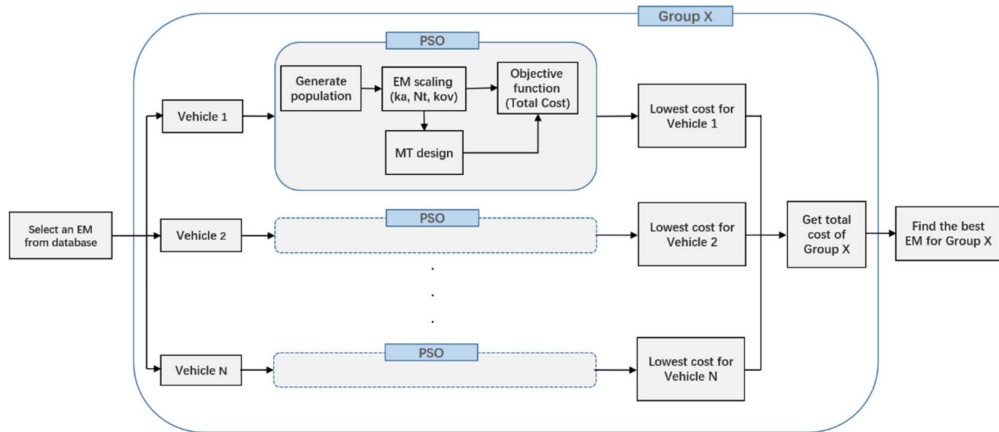

**Figure 2.** Optimization procedure.

### 3.3. Optimization Loop

Particle swarm optimization (PSO) is known to effectively solve large-scale nonlinear optimization problems [10]. The PSO algorithm is used to pick the best solution for a given EM geometry. In this paper, the optimizer has three dimensions (variables): the axial scaling factor $k_a$, the number of turns $N_t$ and the overloading factor $k_{ov}$, which are the inputs for the scaling process. The values of these three variables should respect some boundary conditions, as below.

$$\begin{aligned} &\text{Min } [f(x)], \ x = [k_a \,; \ N_t \,; \ k_{ov}] \\ &\text{s.t} : k_a \ \in \ [0.5, \ 2] \\ &\quad N_t \ \in \ \mathbb{N} \\ &\quad k_{ov} \ \in \ [1, \ 2] \end{aligned} \tag{1}$$

In the above formula, $f(x)$ represents the objective function, as mentioned before, and x is the vector of scaling factors. Before initiating the optimization, the search space will be predefined by (1) and each iteration is defined by Equations (2) and (3) [10], where $\omega$, $C_1$ and $C_2$ are used to make the particles move to control the algorithm's performance.

$$V_i(t) = \omega{\cdot}V_i(t-1) + C_1{\cdot}rand{\cdot}(X_{best,i} - X_i(t-1)) + C_2{\cdot}rand{\cdot}(X_{best} - X_i(t-1)) \tag{2}$$

$$X_i(t) = X_i(t-1) + V_i(t) \tag{3}$$

Every time a particle changes its position in the search space (i.e., the values of the three scaling factors), a scaled EM is generated together with the corresponding PEC and MT, and the new set of gear ratios is optimized to minimize the energy consumption. The value of the objective function is also calculated, estimating the total cost. This process is repeated until the optimization converges to a minimum or the maximum iteration number is reached.

All the optimizer settings and design variables of the optimization loops used in this paper are listed in Table 2.

**Table 2.** Optimizer setting and design variables.

| Optimizer Setting | |
|---|---|
| w | 0.73 |
| $C_1$ | 1.5 |
| $C_2$ | 1.5 |
| $V_{limit}$ | $k_a$: [−0.1, 0.1] $N_t$: [−1, 1] $k_{ov}$: [−0.1, 0.1] |
| $N_{iteration}$ | 50 |
| $N_{particle}$ | 100 |
| **Design Variables** | |
| Boundaries for scaling factors | $0.5 < k_a < 2$ $0 < N_t < \text{round}\left(2{\cdot}m_{max}{\cdot}U_{dc}/\left(\sqrt{2}{\cdot}U_{s,max}\right)\right)$ $1 < k_{ov} < 2$ |
| Constraints | $t_{ov} > 60 \text{ s}$ $P_{em_{nom}} > P_{req_{nom}}$ $P_{em_{max}} > P_{req_{max}}$ |

## 4. Models

### 4.1. Electrical Machines

The EM is one key component of electrified vehicles. It converts electrical to mechanical energy when driving and, in turn, when braking energy recovery happens. In this paper, the V-shape Interior Permanent Magnet Synchronous Machine (VIPMSM) is considered, as it is currently one of the most popular in the automotive industry due to its high power density and good efficiency performance. Additionally, all machines considered in this study feature hairpin windings [11,12], and are cooled by an external water jacket built into the housing. An example of VIPMSM with hairpin windings can be seen in Figure 3, in which the main geometric parameters defining the machines in the set are also illustrated.

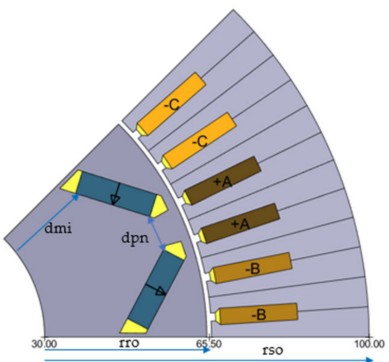

**Figure 3.** V-shape IPMSM geometry.

In order to speed up the optimization process, a predefined set of electrical machine geometries are calculated beforehand, and the results are stored in a database, serving as input to the optimizer. The machines in the set should cover the entire design space of interest, given that the target applications are the different vehicles in the fleet to be optimized. For this reason, a number of geometrical parameters are swept (see Table 3).

The parameters varied to generate the EM database are the stator outer radius ($r_{so}$), the ratio between rotor and stator outer radius ($r_{ro}/r_{so}$), the number of poles ($N_p$), the number of slots per pole and phase (q), the ratio between yoke length and slot length ($K_{iron}$), the distance between v-magnet-slot edge and rotor inner radius (dmi), the distance between the v-magnet-slot edges of the neighbor pole magnets (dpn).

**Table 3.** Parameter ranges considered to create EM database.

| Parameter | Values | Unit |
|:---:|:---:|:---:|
| $r_{so}$ | [80; 100; 120; 140; 160; 180] | mm |
| $r_{ro}/r_{so}$ | [0.6; 0.65] | - |
| $N_p$ | [6; 8; 10; 12; 14] | - |
| $q$ | [2; 3] | - |
| $K_{iron}$ | [0.9; 0.95; 1; 1.05] | - |
| $dmi$ | [0.4; 0.5; 0.6] $(r_{ro} - r_{ri})$ | mm |
| $dpn$ | [0.08; 0.1; 0.12] $(2\pi r_{ro}/N_p)$ | mm |

The parameters in Table 3 give a total of 1584 combinations. Each of these machine geometries are modeled using finite element analysis in the open-source tool FEMM. For each EM geometry, 11 different rotor positions and 25 current combinations are considered. The raw data generated by the FE software is then postprocessed, applying the maximum torque per ampere (MTPA) strategy [13], which ensures maximum torque generation and minimum copper loss in the stator winding, to derive the optimal current combination for each operation point, considering voltage and speed limitations.

The iron losses ($P_{Fe}$) can be estimated by using (4) and then multiplied by a correction factor $k_{cf}$ in order to account for manufacturing effects on iron losses [14]. $c_{hyst}$, $c_{eddy}$ and $c_{ex}$ are constants that can be derived from the material datasheet, $m_{Fe}$ is the iron mass and B is the magnetic flux density in the material.

$$P_{Fe} = \left( c_{hyst} \cdot B^2 \cdot f + c_{eddy} \cdot (B \cdot f)^2 + c_{ex} \cdot (B \cdot f)^{1.5} \right) m_{Fe} k_{cf} \tag{4}$$

The DC copper losses in the stator windings correspond to the resistive losses in the conductors. $P_{DC}$ is determined with (5) and (6) [14] where $\rho_{Cu}$ is the resistivity of copper and $l_{Cond}$ is the total length of the conductors. The slot area, $A_{slot}$, and the copper fill factor, $K_{fill}$, which is set as 0.7 in this paper, define the cross-section area of the conductors. $i_d$ and $i_q$ are the d- and q- components of the stator current respectively.

$$R_{s,DC} = \rho_{Cu} \frac{l_{Cond}}{A_{slot} K_{fill}} \tag{5}$$

$$P_{DC} = R_{s,DC} \left( i_d^2 + i_q^2 \right) \tag{6}$$

For EM with hairpin windings, the AC losses in the windings cannot be neglected as they can be significantly increased due to skin and proximity effects. A major part of these AC losses is primarily caused by the current induced by time-varying leakage flux in the slots [15]. Then, the AC losses in one conductor can be estimated analytically by a homogenized approach [14,16] according to (7)

$$P_{AC} = \frac{1}{24} \sigma \cdot H^3 \cdot W \cdot hm \cdot \sum_{m=1,3,5,\dots}^{\infty} B_m^2 \cdot (m\omega_e)^2 \tag{7}$$

where H and W are the height and width of one conductor, hm is the active length of the electrical machine, σ is the conductivity of the conductor, $B_m$ and $\omega_e$ are the amplitude of the m-order harmonics for the leakage flux and electrical angular velocity of the first order harmonics of the leakage flux for this conductor, respectively. Finally, AC & DC copper losses for the hairpin windings and iron losses in the stator are evaluated, resulting on torque-speed dependent efficiency maps.

Scaling is used to get an accurate and computationally effective way of generating new machines based on the machines from the database. Three factors are introduced to estimate the performance and characteristics of the scaled machines: $k_a$, the ratio between

the scaled EM length and the base one (0.2 m); $N_t$, the desired number of turns; and $k_{ov}$, the ratio between peak and nominal current.

Meanwhile, a lumped parameter thermal model is used to ensure that the thermal constraints are satisfied during the scaling process. Based on this, the overloading factor ($k_{ov}$) is introduced and limits the amount of time that the EM can stay in peak overloading condition. The thermal model used in this paper, shown in Figure 4, is an adaption of the one presented in [5]. The model automatically adjusts itself to any feasible number of conductors per slot. Assuming that there are six conductors in one slot, the thermal model consists of 15 nodes which correspond to the temperature in the outer case, the stator yoke and teeth, the conductors inside the slot and the end turns, the rotor magnets, shaft and bearings. The thermal resistances and capacitances are estimated based on the geometrical parameters [17]. The iron losses for the stator yoke and teeth, the copper losses for each conductor in the slot and each end turn are used as inputs for the corresponding nodes. The maximum allowed winding temperature (150 °C), the coolant temperature (65 °C) and the heat transfer coefficient for the cooling system (600 W/(m²K)) are common to all machines in the set.

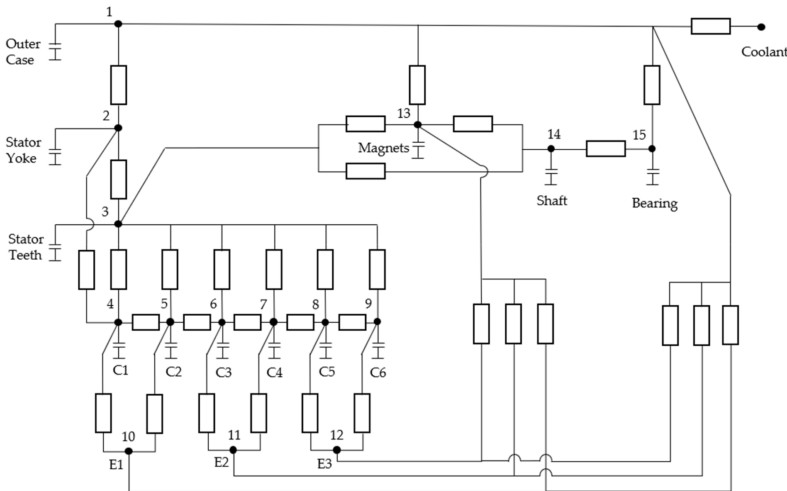

**Figure 4.** Thermal lumped parameter model for EM.

After an EM has been scaled and the new nominal current density has been determined by the thermal model, the corresponding new efficiency map is generated to be used in the powertrain optimization loop.

The upfront cost of the electrical machine is determined by the cost of the materials used and the cost of the major manufacturing operations required to generate and assemble the physical parts, as described in [18,19]. This results in production volumes affecting the upfront cost of the machines, and thus lower costs may be obtained by sharing either complete 3D machine designs or even 2D geometries, due to economies of scale.

### 4.2. Power Electronics Converter (PEC)

The performance model of the PEC is not considered in this work; instead, a fixed efficiency value of 0.97 is assumed when estimating the overall powertrain efficiency. The inverter DC voltage is set to 750 V for all vehicles in the fleet. The cost of the PEC is estimated by (8) based on the work presented in [20], where $PEC_{kVA}$ is the apparent power rating of the PEC.

$$PEC_{cost}(€) = 5.3 PEC_{kVA} + 103 \tag{8}$$

### 4.3. Mechanical Transmission (MT)

In this paper, three types of transmission are considered for each vehicle: single-speed, 2-speed and 3-speed. This not only facilitates using the same machine geometry

for different vehicles, but also provides a better insight into the performance and cost implications of the different design alternatives [21]. Moreover, the selection of gear ratios is optimized to minimize the energy consumption. No matter how many gears an MT has, it should meet the maximum torque and speed requirements arising from the vehicle's specifications.

For a single-speed transmission, given that the selected EM is able to deliver the required power, the gear ratio should be constrained as shown in (9), by the torque and speed requirements.

$$T_{max,wheel} / T_{max,em} \leq g_r \leq n_{max,em} / n_{max,wheel} \tag{9}$$

For a 2-speed transmission, the gear ratios can be constrained as shown in Equations (10) and (11). Moreover, it should be ensured that there is sufficient overlapping between different gears [22], which is shown in (12),

$$gr_1 \geq T_{max,wheel} / T_{max,em} \tag{10}$$

$$gr_2 \leq n_{max,em} / n_{max,wheel} \tag{11}$$

$$n_{max,em} / gr_1 > n_{base,em} / gr_2 \tag{12}$$

For a 3-speed transmission, the gear ratios can be constrained as shown in Equations (13)–(17)

$$gr_1 \geq T_{max,wheel} / T_{max,em} \tag{13}$$

$$gr_3 \leq n_{max,em} / n_{max,wheel} \tag{14}$$

$$gr_2 = \sqrt{gr_1 \cdot gr_3} \tag{15}$$

$$n_{max,em} / gr_1 > n_{base,em} / gr_2 \tag{16}$$

$$n_{max_{em}} / gr_2 > n_{base_{em}} / gr_3 \tag{17}$$

For simplicity, and in an attempt to reflect the friction and splashing losses inside different transmissions, constant efficiency values of 0.98, 0.975 and 0.97 for single-speed, 2-speed and 3-speed MT, respectively, are assumed when computing the overall system efficiency for the powertrain. The cost of the mechanical transmission is estimated by Equations (18)–(20), which are derived from the transmission cost data presented in [6].

$$MTsingle_{cost}(\text{€}) = 1.20 T_{input} + 118 \tag{18}$$

$$MT2_{cost}(\text{€}) = 1.41 T_{input} + 133 \tag{19}$$

$$MT3_{cost}(\text{€}) = 1.57 T_{input} + 156 \tag{20}$$

### 4.4. Operating Cost

The operating cost for each vehicle and its corresponding drive cycle is estimated based on the energy consumption.

First, the acceleration of the vehicle over the drive cycle is calculated. Then, this acceleration together with the speed and the road slope specified in the drive cycle is used to calculate the energy consumption by following the vehicle dynamics equations [23] below.

$$P_{wheel}(k) = (ma(k) + mg sin(\theta(k)) + mg C_r cos(\theta(k)) + \frac{1}{2} C_d \rho A_v v^2(k)) v(k) \tag{21}$$

$$P_{Batt}^+(k) = \frac{P_{wheel}(k)}{\eta_{em}(k) \eta_{transmission} \eta_{PEC}} \tag{22}$$

$$P_{Batt}^-(k) = P_{wheel}(k) \eta_{em}(k) \eta_{transmission} \eta_{PEC} \tag{23}$$

$$EC = \sum_{k=0}^{n} \left( P_{Batt}^{+}(k) + P_{Batt}^{-}(k) \right) \Delta t \tag{24}$$

where EC is the energy consumption over the drive cycle, $P_{wheel}(k)$ is the power delivered to the wheels at instant k, $P_{Batt}^{+}(k)$ and $P_{Batt}^{-(k)}$ are the power delivered by the battery at instant k in motor and generation modes respectively, $\eta_x$ is the efficiency of component $x$, $m$ is the mass of the vehicle, $a(k)$ is the vehicle acceleration at instant k, $g$ is the gravity acceleration, $\theta$ is the angle of the road, $C_r$ is the rolling resistance coefficient, $C_d$ is the air-drag coefficient, $\rho$ is the density of air, $A_v$ is the frontal area of the vehicle and $v(k)$ is the vehicle speed at instant k. As for the transmission, the shifting time is neglected, and at each point in time, the gear which provides the highest efficiency for the EM at that speed and torque operating point is selected. Although taking this as the shifting method is unrealistic, since in some cases it could lead to a very high number of gearshifts in a short time, it still offers a sufficiently accurate estimation for the energy consumption at a reasonable computational time. Finally, the operating cost is estimated by the product of the total annual distance travelled, the energy consumption and the energy price.

## 5. Results

In this section, the proposed methodology and models are used in the optimization of the powertrain for a commercial electric trucks fleet. For each group, there might be some EMs from the database not suited for the specific applications, which will then be discarded by the optimizer.

To illustrate the differences among single-speed, 2- and 3-speed transmission, vehicle 4 is taken as an example. The upfront cost, operating cost and total cost of vehicle 4 for each viable EM are shown in Figure 5.

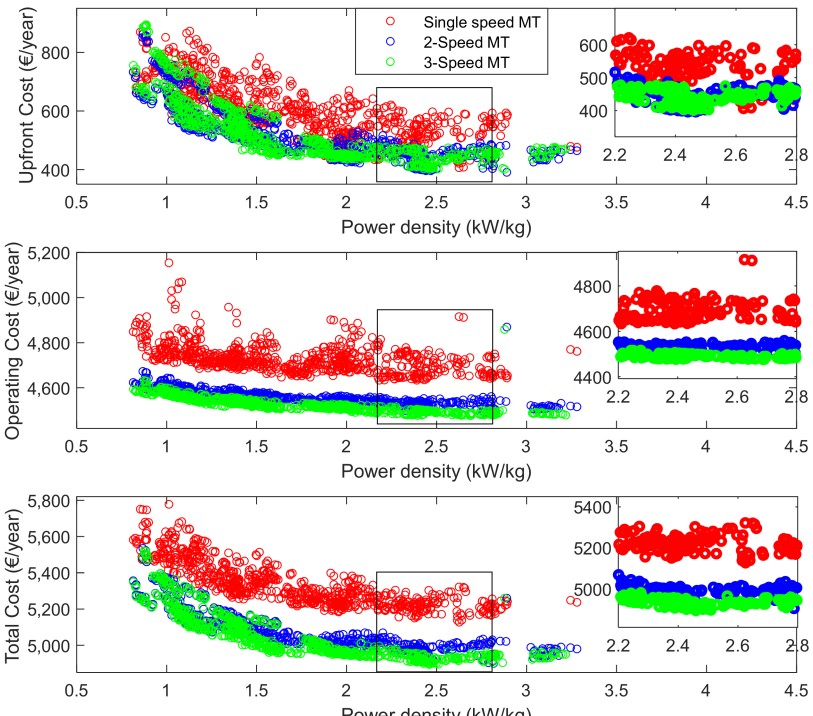

**Figure 5.** Upfront cost (depreciated in 10 years) (top), Operating cost (middle) and total annual cost (bottom) for Vehicle 4 in Group B after optimizing 1584 electrical machine 2D geometries.

It can be observed that the upfront cost, operating cost and total cost in one year for a 21-ton truck (vehicle 4) tend to increase with a single-speed transmission. For the upfront cost, this is due to the fact that a larger EM is needed to meet both the torque and

speed requirements with a single-speed MT, compared with 2 and 3-speed MTs, which leads to higher EM and PEC costs. A multi-speed MT solves the primary issue related to single-speed transmissions: contradictory requirements for high efficiency at top speeds and increased torque at launch and low speeds.

From the results in Figure 5, it is clear that, compared to a single-speed MT, multi-speed MTs offer 4–5% improvement in energy consumption (lower operating cost). Furthermore, a 3-speed MT also provides operating cost benefits compared to a 2-speed MT. To minimize the operating cost, the gear ratios for 2 and 3-speed MT are selected so that the operating points at which most energy is consumed are shifted into the high efficiency area of the electrical machine, as seen in Figure 6a,b. A 3-speed transmission expands the range of gear ratios that can be chosen, allowing for more operating points to be located in the high efficiency area.

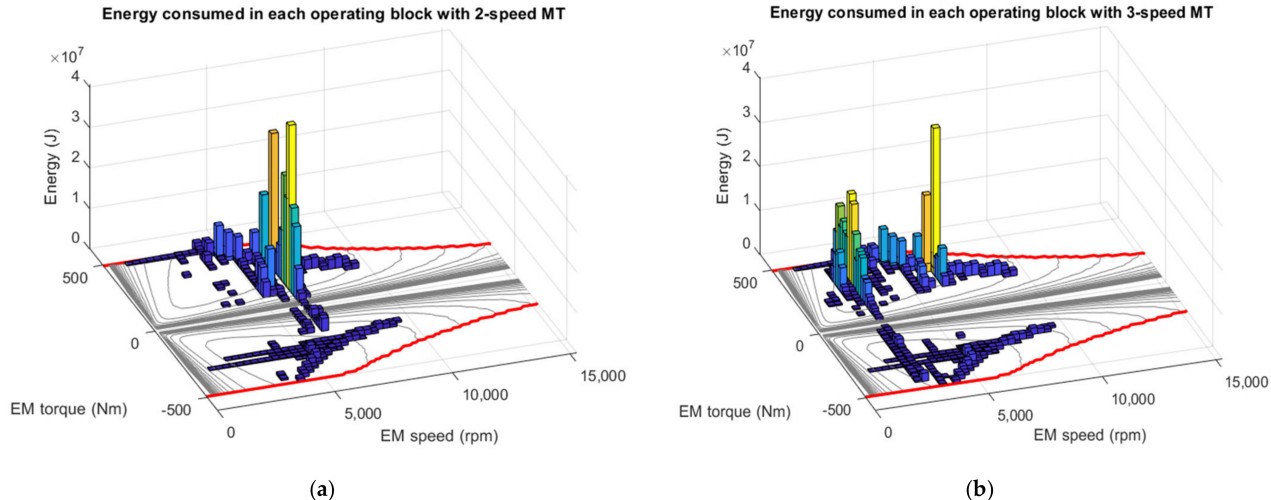

(**a**)                                        (**b**)

**Figure 6.** Each bar represents the energy consumed at that specific operating point with a 2-speed MT (**a**); and with a 3-speed MT (**b**).

The overall total cost for each group is finally obtained by picking the lowest total cost among the single-speed, 2- and 3-speed MT alternatives for each vehicle and adding up the cost for all the vehicles in the group, as explained in Section 3.

In the first considered case, each type of vehicle has a purposely optimized power-train, with a specific EM, PEC and MT tailored to minimize the total cost. The results for all groups are compiled in Table 4.

In Table 4 it can be seen that as the weight of the truck increases, the required power grows, and this leads to optimal EMs having longer active length and larger outer stator radius.

The operating costs vs. the upfront costs for all suitable EM designs for Group B are illustrated in Figure 7. It is clear that the upfront cost of the optimal design for each truck increases from vehicle 4 to 8 because of the increase in power demand. It is also shown that the cheapest machine (lowest upfront cost) is not always the one with the lowest total cost, and the optimizer finds good balance between upfront cost and operating cost.

**Table 4.** Optimal powertrain design for each vehicle type (Case 1).

| | Group A | | | | Group B | | | | Group C |
|---|---|---|---|---|---|---|---|---|---|
| | Veh 1 | Veh 2 | Veh 3 | Veh 4 | Veh 5 | Veh 6 | Veh 7 | Veh 8 | Veh 9 |
| $D_{so}$ (mm) | 200 | 200 | 240 | 240 | 240 | 240 | 240 | 240 | 280 |
| Active length (mm) | 226 | 249 | 200 | 275 | 287 | 343 | 377 | 379 | 345 |
| $N_t$ | 2 | 2 | 2 | 1 | 1 | 1 | 1 | 1 | 1 |
| $N_{conductor}$ | 6 | 6 | 8 | 6 | 6 | 8 | 8 | 8 | 8 |
| kov | 1.34 | 1.36 | 1.67 | 1.66 | 1.60 | 1.43 | 1.41 | 1.52 | 1.43 |
| Np | 6 | 6 | 8 | 8 | 8 | 10 | 8 | 8 | 10 |
| q | 3 | 3 | 2 | 3 | 3 | 2 | 2 | 2 | 2 |
| Kiron | 1.05 | 1 | 1.05 | 1 | 1 | 1 | 1.05 | 1.05 | 1.05 |
| dmi/dpn | 0.5/0.12 | 0.5/0.12 | 0.5/0.08 | 0.5/0.08 | 0.5/0.1 | 0.5/0.08 | 0.5/0.08 | 0.5/0.08 | 0.6/0.1 |
| Pnom (kW) | 125 | 137 | 168 | 204 | 217 | 270 | 314 | 314 | 304 |
| Pmax (kW) | 164 | 176 | 248 | 307 | 325 | 363 | 425 | 443 | 410 |
| Tnom (N) | 189 | 222 | 244 | 287 | 334 | 401 | 264 | 430 | 548 |
| Tmax (N) | 259 | 306 | 424 | 494 | 548 | 588 | 422 | 682 | 823 |
| Max speed (krpm) | 17.6 | 17.6 | 14.7 | 14.7 | 14.7 | 14.4 | 14.7 | 14.7 | 12.6 |

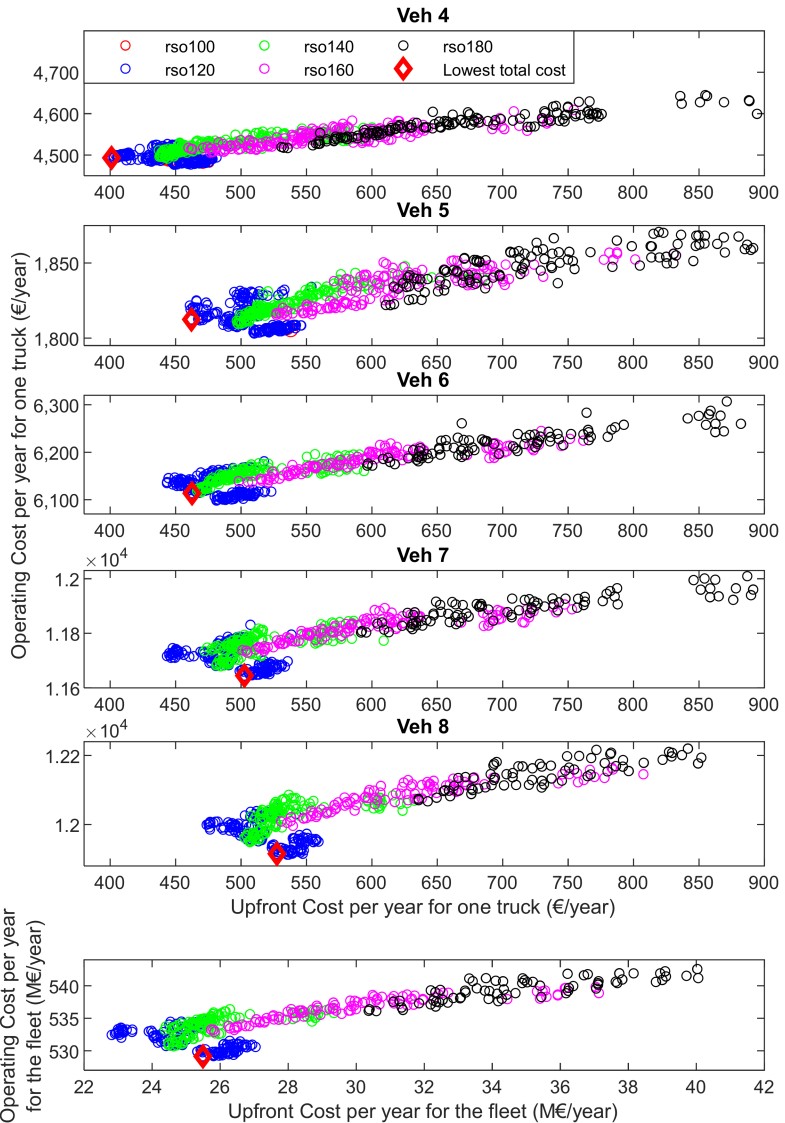

**Figure 7.** Upfront cost and Operating cost for different vehicles in Group B with all EM 2D geometries.

In case 2 all vehicles in the group share the same EM 2D geometry but have an individually optimized active length and overloading factor. Moreover, the PEC and the



MT are also individually sized. In this case, the optimizer will pick the 2D geometry that results in the lowest total cost for the whole group.

Looking again at group B, the optimal 2D geometry for the whole group happens to be the same as the one for Vehicles 7 and 8 in the previous case, which can be seen in the last subfigure in Figure 7. In this case, all the vehicles in the group share the same 2D EM geometry but have individually optimized active length and overloading factor.

In case 3, all vehicles in the group share the exact same EM (2D and 3D), which corresponds to the design from the first case that fulfils the requirements for all vehicles in the group. Looking at Group B again, it can be seen from Table 4 that the EM for vehicle 8 has the highest power, thus meeting the requirements for all other vehicle types. However, this results in oversized machines for many of the other vehicles.

Table 5 summarizes the EM cost, upfront cost of the powertrain, operating cost and total cost of the three groups in three cases.

**Table 5.** EM cost, upfront cost of the powertrain, operating cost and total cost for the three considered cases.

| Case | EM Cost/Upfront Cost/ Operating Cost/Total Cost for Group A (M€/year) | EM Cost/Upfront Cost/ Operating Cost/Total Cost for Group B (M€/year) | EM Cost/Upfront Cost/ Operating Cost/Total Cost for Group C (M€/year) |
|---|---|---|---|
| Case 1 | 2.48/7.38/71.79/79.17 | 8.25/25.19/529.30/554.49 | 2.10/6.41/107.69/114.10 |
| Case 2 | 2.50/7.54/71.74/79.28 | 8.43/25.49/529.14/554.63 | 2.10/6.41/107.69/114.10 |
| Case 3 | 2.59/7.50/71.46/78.96 | 8.27/25.21/529.20/554.41 | 2.10/6.41/107.69/114.10 |

For Groups A and B in cases 2 and 3, even if many vehicles feature an oversized EM, the EM cost does not increase significantly since it benefits from a larger production volume.

Taking Group B again as an example, Figure 8a shows that the EM cost for case 3 (red circle) is lower than the EM cost of the optimized EM for Vehicles 5, 6 and 8 as they have smaller production volumes (even if Vehicle 8 has the largest EM), but it is higher than the cost of the optimized machines for Vehicles 4 and 7 since the increase in production volume does not compensate for the larger size of the machine. From Figure 8b, it can be seen that sharing the same 2D geometry within a group generally increases the EM cost, with the exception of Vehicle 6, in which the EM cost is reduced but the operating cost probably increases to a larger extent. Sharing the same 2D geometry brings limited production cost benefits, which in most cases do not compensate for the larger size resulting from a suboptimal machine design.

Looking at the operating cost, there are only little differences between the three cases. This is due to all EMs selected by the optimizer having good efficiency performance, which, together with optimized gear ratios lead, to almost the same operating cost. The cross-sectional geometry and the efficiency maps of the chosen EMs for vehicle 4 in the three cases considered are shown in Figure 9. The EM in case 1 has 3 slots per pole and phase while the other two EMs only have 2. Furthermore, the EMs in case 2 and 3 have the same 2D geometry but have different active axial lengths due to the different power ratings. All EMs have the same highest efficiency, namely 97%, but the high efficiency area of the EM in case 1 is narrower than that in cases 2 and 3 due to higher copper losses. Nonetheless, the energy consumption for vehicle 4 within different cases is almost the same as a result of the optimized MT, squeezing the operating points into the high efficiency area, which is clearly illustrated in Figure 6.

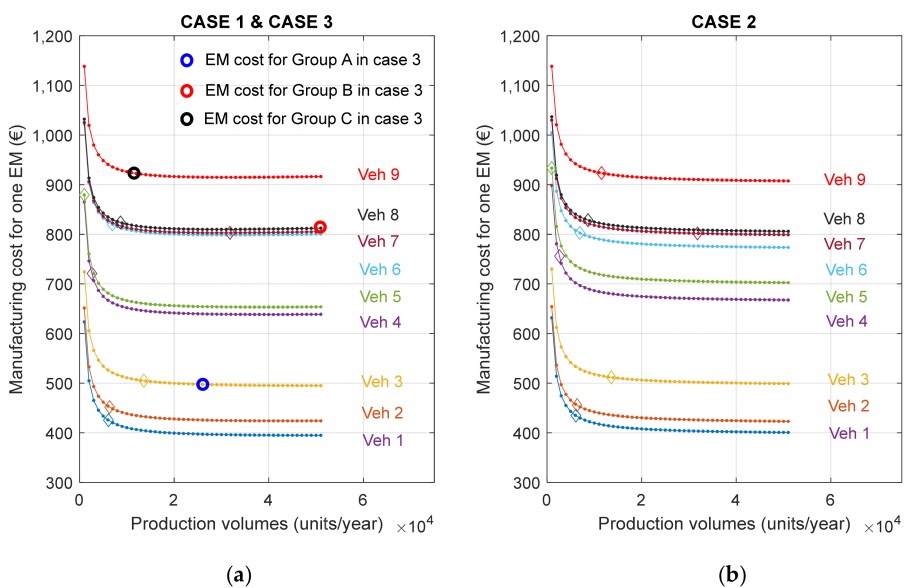

**Figure 8.** (**a**) Manufacturing cost vs. Production volumes in case 1 and case 3; (**b**) Manufacturing cost vs. Production volumes in case 2.

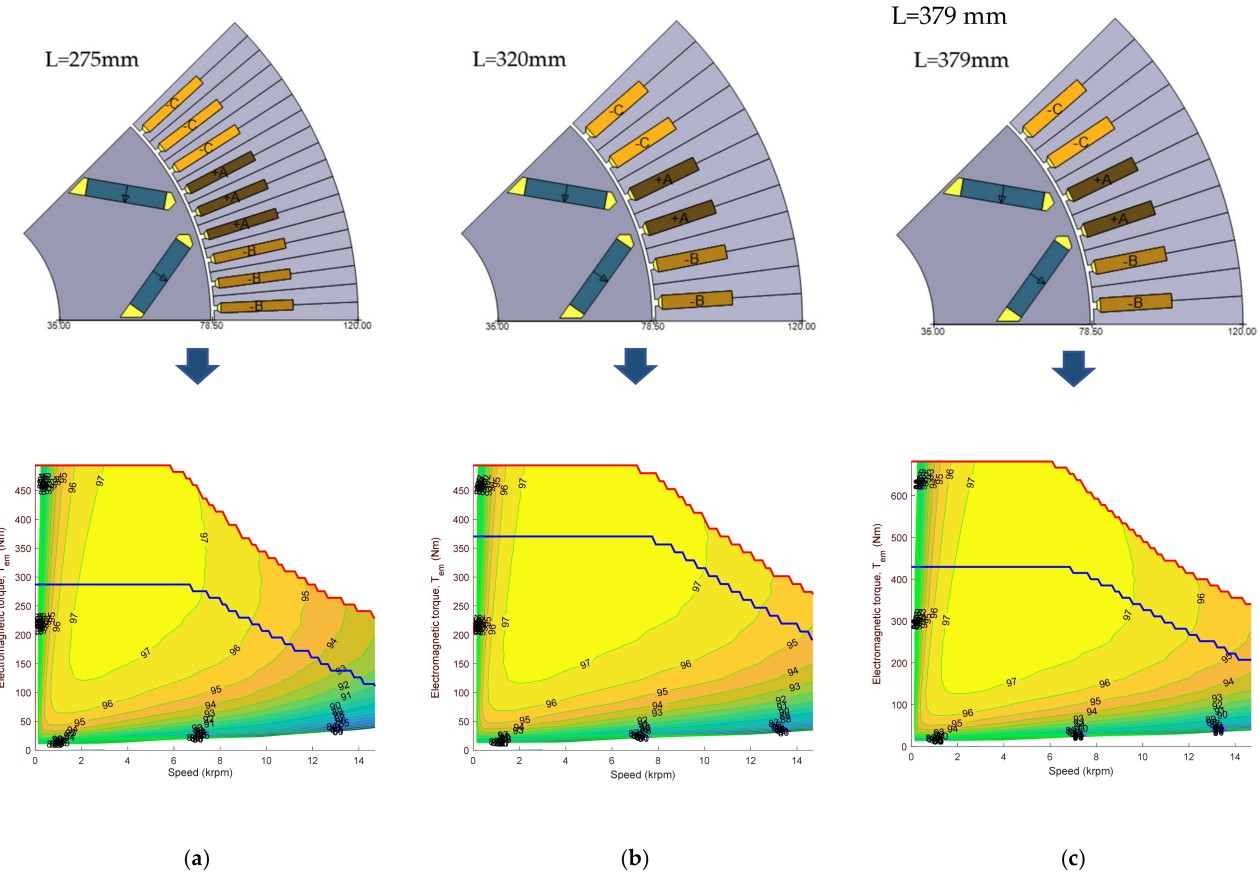

**Figure 9.** 2D EM geometry and the corresponding efficiency map for vehicle 4 in case 1 (**a**), in case 2 (**b**) and in case 3 (**c**).

## 6. Discussion

This paper presents an approach to optimize the powertrain of electric commercial vehicles based on fleet level targets, which is rarely explored, by using scalable performance

models and detailed cost models for the electrical machine, and first order cost models for the power electronics converter and the mechanical transmission. This method provides a comprehensive way to design the powertrain while maximizing potential synergies between applications. In this work, nine different vehicles have been analyzed under three different cases: (i) using a specifically optimized powertrain for each; (ii) sharing the same 2D electrical machine geometry, with individually optimized active length and number of turns, as well as the power converter and transmission layout; and (iii) one single electrical machine for all vehicles in each group.

The presented methodology allows quantifying the cost implications of sharing the electrical machine design, partially or totally, among different vehicles in the fleet. This has direct implications on the electrical machine manufacturing cost and associated logistics, by increasing the synergies between different vehicles.

The preliminary results show that even if the total cost penalty incurred when sharing the same electrical machine is small, the lowest total cost is obtained in case 1, with purposedly optimized powertrains for each vehicle type. However, the presented results do not take into account the cost associated with the development, testing and validation of a new electrical machine design, which—according to previous industry experience—can vary between few million euros for slight changes all the way up to several tens of millions for completely new designs. Thus, taking these costs into consideration will increase the cost saving potential of a higher degree of electrical machine commonality within the fleet.

Although the methodology presented in this paper is supported by similar practices emerging in the automotive industry, where platform-based powertrains are developed for different vehicle classes, the results obtained are limited by the initial assumptions made (such as considering a powertrain with a single traction motor, the motor topology or the gear box configuration). If these assumptions are changed, the method still holds, but new explorations will be needed in order to evaluate the potential benefits of different degrees of powertrain commonality.

In the future, detailed component models for both the power electronics converter and the mechanical transmission will be introduced in order to better represent the powertrain efficiency. The cost models for these components will also be improved in order to account for the possibility of increasing the degree of commonality for these components as well, sharing both design details and production facilities among different applications.

**Author Contributions:** Conceptualization, M.A.; Data curation, M.A.; Funding acquisition, G.D.-O., P.F. and M.A.; Methodology, M.L., G.D.-O., F.J.M.-F., P.F. and M.A.; Project administration, G.D.-O.; Software, M.L., G.D.-O. and P.F.; Supervision, G.D.-O., F.J.M.-F. and M.A.; Validation, M.L. and F.J.M.-F.; Visualization, M.L.; Writing—original draft, M.L.; Writing—review & editing, G.D.-O., F.J.M.-F. and M.A. All authors have read and agreed to the published version of the manuscript.

**Funding:** This research was funded by Swedish Energy Agency, grant number 50213-1.

**Conflicts of Interest:** The authors declare no conflict of interest.

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
