# Peer review of "Electric Drivetrain Optimization for a Commercial Fleet with Different Degrees of Electrical Machine Commonality"

_energies, doi:10.3390/en14112989_

Round 1

Reviewer 1 Report

The powertrain of electric commercial vehicles was optimized based on fleet level targets by using scalable performance models and detailed cost models for the electrical machine.The manuscript has a clear description of the problem, and the experimental comparison is reasonable. The overall quality of the manuscript is good. The following points need to be corrected:
(1) Are the conclusions of this article universal and practical? If the vehicle's power system and transmission system change (such as front-drive, rear-drive, series, parallel, hybrid, etc. HEV), do the conclusions of this article still hold?
(2) What are the shortcomings of existing research? Compared with the existing research, where is the innovation of this article?
(3) Are there actual models/vehicles to verify the research results or conclusions of this article?

Author Response

Response to Reviewer 1:
Thanks for your comments on our paper.

  1. Are the conclusions of this article universal and practical? If the vehicle’s power system and transmission system change (such as front-drive, rear-drive, series, parallel, hybrid, etc. HEV ), do the conclusions of this article still hold?
    Response: We thank the reviewer for raising this point, the following sentences have been added to the revised manuscript.

Line 414:  ‘‘Although the methodology presented in this paper is supported by similar practices emerging in the automotive industry, where platform-based powertrains are developed for different vehicle classes, the results obtained are limited by the initial assumptions made (such as considering a powertrain with a single traction motor, the motor topology, or the gear box configuration). If these assumptions are changed, the method still holds, but new explorations will be needed in order to evaluate the potential benefits of different degrees of powertrain commonality. ‘’

  1. What are the shortcomings of existing research? Compared with the existing research, where is the innovation of this article?
    Response:  The existing research focuses more on optimizing the powertrain for one single vehicle. However, the optimization methodology presented in this article is addressed from a fleet perspective, which is rarely explored.

Line 393: ‘’which is rarely explored’’ added to stress the shortcoming of existing research.

  1. Are there actual models/vehicles to verify the research results or conclusions of this article?

Response:   Both the performance models and the cost models for the different powertrain components (power electronic converters, electrical machines and mechanical transmission) are based on either own work or existing literature. In all cases, these component models have been validated with experimental and/or industrial experience.

As for the methodology, it is difficult to validate the results as the study object is the complete commercial vehicle fleet in Sweden – only time would tell if the full fleet becomes electric, and how the different vehicles composing the fleet look like. However, several companies in the automotive industry apply similar principles, developing platform-based powertrains with different power levels for different vehicle classes. As examples, BorgWarner has launched the iDM product line, with peak power levels of 90 kW, 120 kW and 160 kW for different passenger vehicle sizes. On the other hand, BYD offers its e-Platform with powertrains featuring 45 kW for 1 ton vehicle, 70 kW for 1.2 – 1.6 ton, 120 kW for 1.7 – 2.2 ton and 180 kW for 2.3 – 2.6 ton vehicles.

Reviewer 2 Report

The paper is well written in general. Nevertheless, there are some typing errors that should be improved before publication.

References to sources of the equations should be added.

Figure 9 should be discussed in more detail, and the shown characteristics and the differences in the variants should be elaborated.

Author Response

Thanks for your comments on our paper. We have revised our paper according to your comments:

  1. The paper is well written in general. Nevertheless, there are some typing errors that should be improved before publication.

Response: Thanks for the suggestions. These are now corrected in the revised manuscript.

  1. References to sources of the equations should be added.

Response: Thanks for the suggestions. These are now corrected in the revised manuscript.  

Line 130: reference [10] is added for Equations (2,3).

Line 181: reference [14] is added for Equations (5,6).

Line 274: reference [23] is added for Equations (21-24).

  1. Figure 9 should be discussed in more detail, and the shown characteristics and the differences in the variants should be elaborated.

Response:  We have followed the suggestion by the reviewer.  

Line 377: “The EM in case 1 has 3 slots per pole and phase while the other two EMs only have 2. Furthermore, the EMs in case 2 and 3 have the same 2D geometry but have different active axial length due to the different power rating. All EMs have the same highest efficiency, namely 97%, but the high efficiency area of the EM in case 1 is narrower than that in cases 2 and 3 due to higher copper losses. Nonetheless, the energy consumption for vehicle 4 within different cases is almost the same as a result of the optimized MT, squeezing the operating points into the high efficiency area, which is clearly illustrated in Figure 6.”

Line 385: active lengths of the resulting EMs in the three cases are added.

Reviewer 3 Report

The topic of the thesis is interesting.
And it covers advances in terms of electric drivetrain optimization.
The authors are based on a scalable electromechanical model and used by a swarm of particles as the main optimization algorithm.
The results show that the total cost penalty incurred when sharing the same electric machine is less, so electric machine commonality has the potential for cost savings at a higher level and is considered suitable as a paper.
The proposed method was supported by the results of implementation, and further research plans are expected.

Author Response

Thanks for your comments on our paper.